# Comparative Stability Study of Polysorbate 20 and Polysorbate 80 Related to Oxidative Degradation

**DOI:** 10.3390/pharmaceutics15092332

**Published:** 2023-09-16

**Authors:** Benedykt Kozuch, Johanna Weber, Julia Buske, Karsten Mäder, Patrick Garidel, Tim Diederichs

**Affiliations:** 1PDB-TIP, Innovation Unit, Boehringer Ingelheim Pharma GmbH & Co. KG, Birkendorfer Straße 65, 88397 Biberach an der Riss, Germany; 2Institute of Pharmacy, Faculty of Biosciences, Martin-Luther-University Halle-Wittenberg, Wolfgang-Langenbeck-Strasse 4, 06120 Halle, Germany

**Keywords:** biotherapeutic formulations, surfactants, polysorbates, polysorbate stability, Tween^®^, autoxidation, oxidative degradation

## Abstract

The surfactants polysorbate 20 (PS20) and polysorbate 80 (PS80) are utilized to stabilize protein drugs. However, concerns have been raised regarding the degradation of PSs in biologics and the potential impact on product quality. Oxidation has been identified as a prevalent degradation mechanism under pharmaceutically relevant conditions. So far, a systematic stability comparison of both PSs under pharmaceutically relevant conditions has not been conducted and little is known about the dependence of oxidation on PS concentration. Here, we conducted a comparative stability study to investigate (i) the different oxidative degradation propensities between PS20 and PS80 and (ii) the impact of PS concentration on oxidative degradation. PS20 and PS80 in concentrations ranging from 0.1 mg⋅mL^−1^ to raw material were stored at 5, 25, and 40 °C for 48 weeks in acetate buffer pH 5.5 and water, respectively. We observed a temperature-dependent oxidative degradation of the PSs with strong (40 °C), moderate (25 °C), and weak/no degradation (5 °C). Especially at elevated temperatures such as 40 °C, fast oxidative PS degradation processes were detected. In this case study, a stronger degradation and earlier onset of oxidation was observed for PS80 in comparison to PS20, detected via the fluorescence micelle assay. Additionally, degradation was found to be strongly dependent on PS concentration, with significantly less oxidative processes at higher PS concentrations. Iron impurities, oxygen in the vial headspaces, and the pH values of the formulations were identified as the main contributing factors to accelerate PS oxidation.

## 1. Introduction

Polysorbate 20 (PS20) and polysorbate 80 (PS80) are non-ionic surfactants, which are used in biopharmaceuticals to ensure the stability of monoclonal antibodies/proteins and to prevent protein particle formation in drug products [1,2,3,4]. Due to their low toxicity, high biocompatibility, and excellent stabilization properties they are gold standard excipients in parenteral formulations [5,6]. Nevertheless, in the last decade concerns were raised about the intrinsic stability of polysorbates (PS) [7,8]. PS degradation can be classified as (i) hydrolysis and (ii) oxidation [8,9,10,11]. Hydrolysis can be further subdivided into chemical hydrolysis, which occurs under acidic and basic conditions [10,12] and enzyme-mediated hydrolysis, which is induced by residual host cell proteins [13,14,15,16,17,18,19,20]. In particular, enzyme-mediated hydrolysis is a drastic problem for the pharmaceutical community due to the release of free fatty acids that may form visible particles [21,22]. The second degradation mechanism of PS is oxidation, which has been observed under pharmaceutically relevant formulation conditions [9]. The importance of understanding oxidative PS degradation was highlighted by a survey of 16 globally acting companies, demonstrating that PS degradation was observed through both hydrolysis (69%) and oxidation (63%) in at least one of their biopharmaceutical products [22].

The oxidation process is a free radical chain reaction composed of an initiation, propagation, and termination step [8,9]. Initially, an alkyl radical (R^•^) is generated, followed by the reaction with molecular oxygen to form peroxyl radicals (ROO^•^), which further propagate the radical chain reaction by hydrogen abstraction or disintegration. As long as a radical reacts with a non-radical compound, the chain reaction is propagated [7,10]. Finally, two radical species terminate the chain reaction by reacting with each other. Other possibilities for termination are e.g., (i) lack of oxidizable substrate, (ii) the absence of oxygen [23], or (iii) the presence of antioxidants [10], which scavenge radicals formed during the chain reaction. Many causes of PS oxidation have been discussed, such as heat, light, metal contaminations, or reactive oxygen species (ROS) [8,9,24]. In particular, the exposure to stainless steel or contact with metal ions from raw products have repeatedly been reported to promote PS oxidation [25,26,27,28,29,30,31,32,33]. ROS such as hydroxyl radicals (HO^•^), superoxide radicals (^•^O_2_^−^), peroxyl/hydroperoxyl radicals (ROO^•^/HOO^•^), water peroxide H_2_O_2_, hydroperoxide (ROOH), or singlet oxygen (^1^O_2_) are highly reactive species, commonly found in oxidative processes. In particular, iron (Fe^2+^) is known for its oxidizing properties and can react with H_2_O_2_ via the Fenton reaction to form hydroxyl radicals (OH^•^) or react with oxygen to produce a superoxide radical (^•^O_2_^−^), which can produce a hydroxyl radical via the Haber–Weiss reaction in the presence of H_2_O_2_ [34,35,36,37]. The Fenton reaction is influenced by numerous factors such as the type of metal, the ligand of the metal, the nature and concentration of the substrate, or the pH [38,39,40,41,42,43]. Especially at low pH values of approximately 3, the Fenton reaction is far more likely to occur [44,45,46]. Redox reactions of Fe^2+^ and Fe^3+^ with organic matter, Equation (1), peroxides, Equations (2) and (3), or molecular oxygen, Equation (4), can also initiate radical formation [27,47,48].
(1)$${\mathrm{Fe}}^{3+}+\mathrm{RH}\to {\mathrm{Fe}}^{2+}+R^{•}+H^{+}$$
(2)$${\mathrm{Fe}}^{2+}+\mathrm{ROOH}\to {\mathrm{Fe}}^{3+}+{\mathrm{OH}}^{-}+{\mathrm{RO}}^{•}$$
(3)$${\mathrm{Fe}}^{3+}+\mathrm{ROOH}\to {\mathrm{Fe}}^{2+}+H^{+}+{\mathrm{ROO}}^{•}$$
(4)$${\mathrm{Fe}}^{2+}+O_{2}\to {\mathrm{Fe}}^{3+}+O_{2}^{•-}$$

Some studies investigated the radical formation in PS during oxidation. For instance, Mittag and colleagues (2022) compared the formation of OH^•^, R^•^, RO^•^, and ROO^•^ radicals in PS80 for different vendors and different concentrations [49]. They mainly observed RO^•^ and ROO^•^ radicals relative to each other in bulk PS80 solutions as well as mainly OH^•^ radicals in 10% (*w*/*v*) aqueous solutions [49]. Different radicals differ drastically in their reactivity [50,51,52]; for instance, OH^•^ radicals react in a diffusion-controlled manner, whereas for ROO^•^ or superoxide radicals, the reaction constants are six to nine orders of magnitude lower [49,50,51,52,53].

Comparing the chemical structures of the main components of PS20 and PS80, the latter is generally considered to be more susceptible to oxidative degradation due to a higher content of unsaturated fatty acids [7,24,31,54,55,56,57]. For instance, Yao et al. (2009) reported a 2.65 times higher oxidizability constant for PS80 than for PS20 based on the higher content of oleic acid in PS80 and the energetically favored H-abstraction in the vicinity of the double bond [57]. Potential reactivity sites for oxidation in PS20 and PS80 as well as reported oxidative degradation products are shown in Figure 1. A number of key characteristics in relation to the oxidative degradation of polysorbate were identified, such as (i) pH shifts in unbuffered solutions [9], (ii) preferred degradation of higher-order esters [25,26,54,55,58], (iii) dominant degradation of longer fatty acid esters [55,59], (iv) competition with proteins for oxidation [27,30,54], and (v) emerging species with lower hydrophobicity for PS80 [25,26,31,54,55]. For a more detailed description and summary, see Weber et al. (2023) [60].

Many studies revealed a higher oxidation susceptibility for PS80 in comparison to PS20 [7,24,31,54,55,56,57]. For instance, Kishore and colleagues (2011) reported that thermal oxidation of PS80 in placebo formulations is initiated at the olefinic site and afterwards is mainly degraded at the POE chain [7]. Slightly higher peroxide rates and slightly more degradation were reported for storage at 40 °C for 24 weeks due to the hydrocarbon chain degree of unsaturation, however, the overall degradation profiles were comparable between PS20 and PS80 [7]. Borisov et al. (2015) revealed a higher susceptibility to oxidation for unsaturated fatty acids after stressing PS20 and PS80 formulations in the absence of proteins with the artificial radical initiator 2,2-azobis(2-amidinopropane) dihydrochloride (AAPH) by determining pseudo-first order rates for the degradation of sorbitan POE esterified to different fatty acids with LC-MS [55]. Hvattum and colleagues (2012) observed complete degradation of C18:2 in PS80 placebo formulations after 8 weeks at 40 °C [56]. Based on the higher content of unsaturated fatty acids — oleic and linoleic acid — in PS80 an energetically favored abstraction of a hydrogen atom in the vicinity of a double bond is more likely to occur. As a result, a larger oxidizability constant of 2.65 was reported for PS80 in comparison to PS20 [57]. Additionally, more degradation products or oxidation markers were found in PS80 originating from the vicinity of the double bond, such as esters of 9-oxo-C9:0, [25,55] hydroxyl-C18:1 [25,55,56], hydroperoxyl-C18:1 [25,55], keto-C18:1 [25,55], epoxy-C18:0 [55,56], 1,9-nonanedioic acid [54], or 2-decenedioic acid [31], as summarized in Figure 1C. Although numerous studies have addressed the oxidation of PS20 and PS80, a systematic and PS concentration-dependent comparison between both polysorbates under pharmaceutically relevant conditions (5 °C without oxidation accelerators) is currently missing in the literature [60].

Here, we investigated the oxidative degradation of PS20 and PS80 in placebo formulations for concentrations ranging from 0.1 mg⋅mL^−1^ PS to raw materials in 25 mM acetate buffer pH 5.5 and ultrapure water at 5, 25, and 40 °C. We observed a temperature-dependent degradation, as only little or no degradation was observed at 5 °C, and fast oxidation processes were determined at elevated temperature of 25 °C and 40 °C. Independent of the PS type, a slower degradation was observed for increasing initial PS concentrations. Additionally, a faster onset of oxidation was observed for PS80 placebo formulations via fluorescence micelle assay (FMA), especially at 5 °C.

## 2. Materials and Methods

### 2.1. Materials

PS20 high purity (HP) (Croda Inc., Mill Hall, PA, USA); PS80 HP (Croda Inc., Mill Hall, PA, USA); sodium acetate trihydrate (Merck KGaA, Darmstadt, Germany); acetic acid (Honeywell International Inc., Charlotte, NC, USA); nitric acid (Carl Roth GmbH+ Co. KG, Karlsruhe, Germany); NPN (*N*-phenylnaphthalen-1-amine; Merck KGaA, Darmstadt, Germany); Brij™-35 (polyoxyethylene (23) lauryl ether; Thermo Fisher Scientific, Waltham, MA, USA); NaCl (Merck KGaA, Darmstadt, Germany); acetonitrile (Carl Roth GmbH+ Co. KG, Karlsruhe, Germany); Tris (tris(hydroxymethyl)aminomethane; Merck KGaA, Darmstadt, Germany); ammonium iron(II)sulfate hexahydrate (Merck KGaA, Darmstadt, Germany); sulfuric acid (Merck KGaA, Darmstadt, Germany); sorbitol (Merck KGaA, Darmstadt, Germany); xylenol orange disodium salt (2,2′,2″,2‴-{(1,1-Dioxo-2,1λ6-benzoxathiole-3,3(1*H*)-diyl)bis[(6-hydroxy-5-methyl-3,1-phenylene)methylenenitrilo]}tetraacetic acid disodium salt; Merck KGaA, Darmstadt, Germany); H_2_O_2_ (Merck KGaA, Darmstadt, Germany); methanol (Fisher Scientific GmbH, Schwerte, Germany); sulfuric acid (Merck KGaA, Darmstadt, Germany); formic acid (Merck KGaA, Darmstadt, Germany); ammonium formate (Merck KGaA, Darmstadt, Germany). All tested formulation compounds were analytical grade.

#### Formulation and Storage of the Polysorbate Samples

PS20 HP and PS80 HP at concentrations of 0.1, 0.2, 0.4, 1, 10, 100, and 438 mg⋅mL^−1^ were prepared in 25 mM acetate buffer pH 5.5 and ultrapure water followed by sterile filtration using Millipak^®^ 40 polyvinylidene fluoride (PVDF) filters (Merck KGaA, Darmstadt, Germany). 10 mL Fiolax^®^ clear TOPLINE glass vials (SCHOTT Pharma AG & Co. KGaA, Mainz, Germany) were filled with 5 mL sample volume, stoppered with D777-1 20 mm laminated butyl rubber stoppers (Daikyo Seiko, Ltd., Sano, Japan), and sealed airtight with aluminum crimp caps (West Pharmaceutical Services, Inc., Eschweiler, Germany). The samples were incubated at 5 °C, 25 °C/60% rh, and 40 °C/75% rh, protected from light, and retrieved at selected sampling time points. Additionally, 1 mL samples of PS20 and PS80 raw materials were filled without filtration and stored under the same conditions. Due to the high viscosities, filtration of the raw materials was not possible.

### 2.2. Methods

#### 2.2.1. Inductively Coupled Plasma-Mass Spectrometry (ICP-MS)

For the raw materials of PS20 and PS80 as well as for a solution of 1 M acetate buffer pH 5.5, the metal ion concentrations were determined by ICP-MS, utilizing a 7800 ICP-MS (Agilent, Santa Clara, CA, USA) system equipped with an SPS 4 autosampler (Agilent, Santa Clara, CA, USA). PS20 and PS80 raw material samples were dissolved through acid digestion using nitric acid and measured against custom standards of the metals Al, B, Cr, Cu, Fe, Mn, Mo, Ni, and V. No-gas-mode or helium-mode was used depending on the element. The limit of quantification (LOQ) was dependent on the sample and was 250 ng⋅mL^−1^ for B, 250 ng⋅mL^−1^ for Al, 2.5 ng⋅mL^−1^ for V, 100 ng⋅mL^−1^ for Cr, 10 ng⋅mL^−1^ for Mn, 100 ng⋅mL^−1^ for Fe, 25 ng⋅mL^−1^ for Ni, 10 ng⋅mL^−1^ for Cu, and 10 ng⋅mL^−1^ for Mo for the raw materials of PS20 and PS80, as well as for the 1 M acetate buffer sample. For the 100 mg⋅mL^−1^ samples of PS20 and PS80 the LOQ was 250 ng⋅mL^−1^ for B, 250 ng⋅mL^−1^ for Al, 2.7 ng⋅mL^−1^ for V, 6.57 ng⋅mL^−1^ for Cr, 4.94 ng⋅mL^−1^ for Mn, 10 ng⋅mL^−1^ for Fe, 25 ng⋅mL^−1^ for Ni, 10 ng⋅mL^−1^ for Cu, and 5 ng⋅mL^−1^ for Mo.

#### 2.2.2. pH Measurements

pH measurements were conducted with the samples at room temperature using a SevenCompact S220 pH-meter with an InLab Micro Pro-ISM pH electrode (Mettler Toledo, Columbus, OH, USA). The pH electrode was calibrated with pH 4.01 and pH 7.00 buffer solutions (Mettler Toledo, Columbus, OH, USA).

#### 2.2.3. Fluorescence Micelle Assay (FMA)

The FMA was adapted from Lippold et al. (2017) and Glücklich et al. (2021) [58,61]. FMA measurements were conducted using a Fluent^®^ Automation Workstation (Tecan Group AG, Männedorf, Switzerland). Here, 240 µL of FMA buffer consisting of 5 µM *N*-phenylnaphthalen-1-amine (NPN), 0.0015% (*w*/*v*) Brij™-35, 150 mM NaCl, 5% (*v*/*v*) acetonitrile, and 50 mM Tris at pH 8.0 were added to 10 µL sample volume and incubated at 35 °C for 1 min. The fluorescence was measured with an Infinite M200pro fluorescence plate reader (Tecan Group AG, Männedorf, Switzerland) with an excitation wavelength of 350 nm and an emission wavelength of 420 nm in quadruplicates (*n* = 4). Measurements were conducted against PS20 (0.1–0.6 mg⋅mL^−1^) and PS80 (0.05–0.3 mg⋅mL^−1^) calibration standards. It is worth mentioning that only micelle-forming species are detected and the fluorescence intensity is dependent on the hydrophobicity of the species. Preferential degradation of certain species can result in inaccurate PS concentration determinations. The LOQ was 0.1 mg⋅mL^−1^ for PS20 and 0.05 mg⋅mL^−1^ for PS80.

#### 2.2.4. Ferrous Oxidation with Xylenol Orange (FOX) Assay

FOX measurements were performed to determine the peroxide concentration (every mol ROS oxidizes 1 mol Fe^2+^ to Fe^3+^) in the formulations as described in Jaeger et al. (1994) [62]. The assay was performed using a Fluent^®^ Automation Workstation (Tecan Group AG, Männedorf, Switzerland). In brief, the FOX reagent was prepared by mixing 30 mL of solution A (25 mM ammonium iron(II)sulfate hexahydrate and 2.5 M sulfuric acid) with 300 µL of solution B (100 mM sorbitol and 125 µL xylenol orange disodium salt). For each sample, 30 µL sample and 300 µL FOX reagent were mixed and incubated at RT for 15 min. The absorbance at 595 nm was subsequently measured with a SpectraMax M3 plate reader (Molecular Devices, LLC., San Jose, CA, USA). Measurements were performed in triplicates (*n* = 3) and conducted against H_2_O_2_ calibration standards ranging from 0.39 to 100 µM. The LOQ was 5 µM H_2_O_2_ equivalents (5 µM H_2_O_2_ ≈ 0.17 ppm H_2_O_2_).

#### 2.2.5. Gas Chromatography (GC)

GC measurements were performed to determine the emerging short chain formic acid content of the different formulations with advancing oxidation, as was previously described by Mahadevan et al. (1967) [63]. A 0.5 mL sample was diluted with 0.5 mL methanol and 0.1 mL sulfuric acid was added. The samples were sealed airtight and measured using a 7890A GC system with a polyethylene glycol column (Agilent Technologies, Inc., Santa Clara, CA, USA). Helium was used as carrier gas at 60 kPa. Measurements were conducted against formic acid calibration standards ranging from 20.16 to 504 µmol⋅mol^−1^. The LOQ was 10 µmol⋅mol^−1^ formic acid.

#### 2.2.6. Reversed Phase-Ultra High Performance Liquid Chromatography-Mass Spectrometry (RP-UPLC-MS)

The RP-UPLC-MS method was adapted from Lippold et al. (2017) and Evers et al. (2020) [58,64]. Measurements were conducted using an Ultimate 3000^®^ UHPLC system with RS dual gradient pump, RS autosampler, and RS column compartment (Thermo Fisher Scientific, Waltham, MA, USA) coupled to an ACQUITY QDa mass detector (Waters Corporation, Milford, MA, USA) with an ESI source. For solid-phase extraction and protein removal a mixed-mode column (Waters Oasis^®^Max; 30 µm, 2.1 mm × 20 mm, 80 Å) was used. A Poroshell 120 SB-C8 4.6 × 100 mm, 2.7 μm reversed-phase column (Agilent Technologies, Inc., Santa Clara, CA, USA) was used as the stationary phase for PS subspecies separation. Three mobile phases (A: 100% acetonitrile, B: 100% ultrapure water, and C: 100% methanol) were used for the analytic gradient. After PS subspecies separation by the stationary phase, the analytic gradient, delivered by the right pump (0.7 mL⋅min^−1^), was mixed with a 10 mM ammonium formate buffer, delivered by the left pump (0.2 mL⋅min^−1^), using a T-piece. The injection volume was 2 µL and the column oven was set to 50 °C (PS20) or 60 °C (PS80). The method had a run time of 45 min. The QDa detector analyzed masses between 250–1250 Da in positive mode with a sampling rate of 2 Hz. Measurements were conducted against PS20 and PS80 calibration standards ranging from 0.05 to 0.6 mg⋅mL^−1^ PS. The LOQ was 0.05 mg⋅mL^−1^ for both polysorbates.

#### 2.2.7. Oxygen Measurements

Oxygen concentrations in the headspaces were measured with a Microx 4 trace fiber optic oxygen meter with an IMP-PSt7 microsensor (PreSens Precision Sensing GmbH, Regensburg, Germany) as described in Hipper et al. (2023) [65]. The samples needed to be sealed to ensure accurate measurements. Therefore, the sensor was mounted into a syringe. With the sensor retracted, the rubber stoppers were pierced with the syringe and the sensor was subsequently introduced into the headspace of the sample. The “dry mode” setting was selected for the measurements. The device was calibrated with pure nitrogen, which corresponded to 0% oxygen, and an oxygen–nitrogen mixture with 20.88% oxygen, which corresponded to the atmospheric oxygen concentration.

## 3. Results

### 3.1. FMA Analysis and Reactive Oxygen Species

The degradation of PS20 and PS80 for concentrations ranging from 0.1 mg⋅mL^−1^ to raw materials in acetate (AB) and ultrapure water (UW) was tracked through FMA (Figure 2A,C and Appendix A) and FOX measurements (Figure 2B,D and Appendix A) over a period of 48 weeks (protected from light) to elucidate the concentration-dependence of oxidation. Additionally, this provides a systematic comparison between the oxidation of PS20 and PS80 for various concentrations. Usually, PS80 is expected to experience faster oxidation based on the high content of unsaturated fatty acids [7,24,31,54,55,56,57], however, an in-depth comparison between both polysorbates is currently missing in the literature [60]. FMA measurements were performed to obtain information on PS content. It is worth mentioning that only micelle-forming species are measured by FMA because the dye NPN partitions within the micelles and the hydrophobicity of these micelle “species” affects the fluorescence intensity. It is expected that inaccurate PS contents are measured due to preferential degradation of certain species. Additionally, FOX analysis was performed to monitor the formation of ROS (H_2_O_2_ equivalent concentrations). Together, the progress of oxidation can be evaluated for the different conditions and storage times.

Figure 1A shows the PS content for different initial PS concentrations up to 48 weeks at 40 °C/75% relative humidity (rh). In general, a PS concentration-dependent degradation of both PS types was observed (Figure 2A). The higher the initial PS concentration, the lower the oxidative degradation propensity. Especially for PS concentrations above 10 mg⋅mL^−1^, less degradation was observed for both PSs after 48 weeks of storage. The PS content ranges from ~52% to more than 80% for 438 mg⋅mL^−1^ PS formulations and raw materials after 48 weeks. For low initial PS concentrations, a strong degradation at 40 °C within the first three months was measured, resulting in a more than 80% PS degradation for concentrations ranging from 0.1 to 1.0 mg⋅mL^−1^. Here, a small difference between PS20 and PS80 was detected, as zero fluorescence was measured for PS80 (0.1–1.0 mg⋅mL^−1^), whereas up to 18% of the remaining PS content was determined for PS20. This difference between PS20 and PS80 content increased continually for 0.4, 1.0, and 10 mg⋅mL^−1^ PS, where PS80 revealed faster degradation kinetics (Figure 2A). The 100 mg⋅mL^−1^ PS20 AB formulation showed a slight degradation of ca. 30%, while the PS20 UW and PS80 formulations degraded to a remaining content of 25–40% after 48 weeks at 40 °C/75% rh. The 100 mg⋅mL^−1^ PS20 UW formulation showed the lowest concentration. The 438 mg⋅mL^−1^ and raw material (RM) formulations exhibit a remaining PS content of 53–85% PS for the same storage conditions (48 weeks at 40 °C/75% rh). The degradation of the 438 mg⋅mL^−1^ PS80 formulations was more severe in comparison to PS20 and similarly a difference of ca. 40% was determined for PS80 RM in comparison to PS20 RM. However, this difference must be taken with caution, as the viscous RM had to be diluted more than 1000-fold for content analysis, leading to larger errors in comparison to the lower concentrated aqueous solutions. This is verified by the higher point-to-point variation in the RM samples, especially visible at later timepoints of the PS80 RM (Figure 2A, panel 8). All PS80 concentrations revealed no difference in degradation kinetics and degree, independent of whether they were formulated in UW or AB (Figure 2A, blue and green lines). For PS20 formulations in the lower concentration range (0.1–1.0 mg⋅mL^−1^) and formulated in water, an initial lag phase of 6–8 weeks without degradation was monitored (Figure 2A). For better visualization, only the first 24 weeks of Figure 2A are shown in Appendix A. This lag phase is not observed for the PS20 acetate formulations as well as for both PS80 formulations and disappeared at concentrations higher than 10 mg⋅mL^−1^. Careful consideration is required when determining the precise content of highly degraded PS, as the FMA is an indirect quantification method that detects polysorbate micelles. Accurate quantification cannot be guaranteed for concentrations below 0.05 mg·mL^−1^ (the lowest calibration point).

Complementary to the PS content, the concentrations of H_2_O_2_ equivalents were measured for PS concentrations ranging from 0.1 to 100 mg⋅mL^−1^ after storage at 40 °C/75% rh. A representation of the H_2_O_2_ equivalent concentration per 0.1 mg⋅mL^−1^ PS in the formulations was chosen to allow a relative comparison between the different PS concentrations. The analyses of 438 mg⋅mL^−1^ formulations and of PS RM samples were omitted due to the low degradation observed in FMA measurements. In general, H_2_O_2_ equivalent concentrations in the formulations followed a roughly symmetrical pattern, with content increasing during the first weeks and decreasing between approximately 6 to 18 weeks (Figure 2B). The maximum H_2_O_2_ equivalent concentrations correlate with the values of the strongest PS degradation (steepest slopes) (Figure 2A,B). With increasing initial PS concentrations, lower H_2_O_2_ equivalent concentrations per 0.1 mg⋅mL^−1^ PS were observed. The latter correlates with the reduced content decrease determined by FMA analysis. This confirms that less ROS normalized to 0.1 mg⋅mL^−1^ PS were present at higher PS concentrations, indicating that fewer oxidation processes occurred normalized to concentration. For PS20 UW formulations, the maxima of the H_2_O_2_ equivalent concentrations shifted to later time points for concentrations ranging from 0.1 to 1.0 mg⋅mL^−1^, whereas no shift was monitored for PS20 UW concentrations above 10 mg⋅mL^−1^ (Figure 2A,B, panel 1–5). The described phenomenon correlates to the delayed initiation of oxidation for PS20 UW as observed in the PS content analysis. For both PS80 formulations and for PS20 AB formulations, the concentration of peroxides increased rapidly after 2–6 weeks, and the highest measured values were reached after 4–8 weeks before a subsequent reduction was measured. Only for 100 mg⋅mL^−1^ PS80 formulations were the highest values detected after 36 weeks (Figure 2B, green and blue lines). However, overall values were very low in comparison to the other formulations (0.5 µM per 0.1 mg⋅mL^−1^ PS). The 0.1–0.4 mg⋅mL^−1^ PS20 UW formulations reached the highest measured values of ROS species after 12 weeks, 4–6 weeks after the equivalent PS20 AB formulations. PS20 HP ultrapure water samples (1–100 mg⋅mL^−1^) exhibit the highest H_2_O_2_ equivalent content after 4–8 weeks. For both PSs, a difference in the measured peroxide concentration between both solvents was detected, with the peak peroxide content of the 0.1 mg⋅mL^−1^ PS20 AB formulation corresponding to approx. 50% of the equivalent PS20 UW formulation. This difference is reduced with increasing PS20 concentration, resulting in a difference of 20% at 100 mg⋅mL^−1^. For PS80, a smaller difference between maximal H_2_O_2_ concentrations of UW and AB formulations was observed, with the largest difference of approx. 40% at 0.4 mg⋅mL^−1^.

FMA and FOX results for storage at 25 °C/60% rh (Figure 2C,D) revealed a similar trend as observed at 40 °C/75% rh, showing lower oxidative susceptibility with increasing PS concentration. Nevertheless, slower kinetics were monitored for the oxidative PS degradation, especially for concentrations ranging from 0.1 to 10 mg⋅mL^−1^ (Figure 2C). As the degradation susceptibility decreases with increasing concentration, the differences between 25 and 40 °C also vanish, especially for concentrations at 100 mg⋅mL^−1^ and higher (Figure 2A,C). The onset of the strong oxidative degradation of PS80 formulations (100 mg⋅mL^−1^) at 25 °C was delayed by 36 weeks, resulting in content ranging from 47 to 55% after 48 weeks. Meanwhile, the corresponding PS20 formulations showed only minimal PS degradation, with 81–94% PS remaining. The decreasing temperature stress at 25 °C revealed larger differences in oxidative degradation between PS20 and PS80. Independent of concentration and solvent type, a faster degradation of PS80 was determined, except for the lowest concentration of 0.1 mg⋅mL^−1^, where both PSs show similar degradation kinetics (Figure 2C). At 0.1 and 1.0 mg⋅mL^−1^ of PS80 UW and AB, nearly no fluorescent signal (full degradation) was detected after approximately 8 weeks at 40 °C, revealing an approximately three-times faster oxidation than for the corresponding formulations stored at 25 °C. The delayed oxidation of the PS20 UW formulations at 0.2 and 1.0 mg⋅mL^−1^, which was also observed for 40 °C, is further delayed, starting between 12 to 18 weeks (Figure 2C). Nevertheless, as observed at higher temperatures, degradation kinetics are comparable between PS20 AB and UW (Figure 2C).

The FOX results revealed a similar picture as for 40 °C/75% rh. The maximum concentration of H_2_O_2_ equivalents was detected at later timepoints and correlates with the steepest decay of PS degradation (Figure 2D). With increasing initial concentrations, lower H_2_O_2_ equivalent concentrations normalized to 0.1 mg⋅mL^−1^ were detected. The lag phase of PS20 UW oxidation correlated with a shifted increase in ROS species (Figure 2D, black graph).

PS20 samples stored at 5 °C showed negligible PS content changes after 48 weeks (Appendix A), with the lowest value of 91% remaining PS measured for the 0.1 mg⋅mL^−1^ PS20 AB formulation. However, all PS80 formulations showed degradation of the surfactant material between 36 and 48 weeks, with remaining PS80 contents of 45–66% after storage for 48 weeks under these conditions. At 0.1 to 1.0 mg⋅mL^−1^, AB formulations showed a stronger degradation, while no notable differences were measured at higher concentrations (Appendix A). The content decrease for all PS80 formulations was confirmed by MS detection. A lag phase could be observed for the PS80 UW formulations in comparison to the PS80 AB formulations for concentrations ranging between 0.1 and 1.0 mg⋅mL^−1^, which is reduced for higher concentrations (Appendix A, green and blue graphs).

The absence of or minimal oxidative degradation of the polysorbates at 5 °C was confirmed by the FOX results. For all PS formulations, very low values of H_2_O_2_ equivalent concentrations were detected. Only for the PS80 concentrations ranging from 0.1 to 0.4 mg⋅mL^−1^, a very small increase of up to approximately 3 µM of H_2_O_2_ equivalents was observed, confirming the earlier onset of oxidative PS80 degradation as observed via FMA (Appendix A).

### 3.2. Acidification of PS20 and PS80 Formulations (pH Analysis and Gas Chromatography)

Oxidative processes are known to produce a variety of degradation products such as, for instance, formic acids, which can cause pH shifts. These changes can influence the oxidation in general, as radical initiation reactions such as, for instance, the Fenton reaction, are pH dependent [44,45,46]. Therefore, pH values and the formic acid content were monitored for PS20 and PS80 UW and AB formulations stored at 5, 25, and 40 °C (Figure 3 and Appendix A). Especially for unbuffered formulations, huge pH shifts were observed (Figure 3A,C), which correlate with the degradation of PS.

Figure 3A shows the pH values of the formulations after 48 weeks of storage at 40 °C. The samples formulated with 438 mg⋅mL^−1^ PS80 as well as PS20 RM and PS80 RM samples could not be measured due to their high viscosities. All samples stored at 40 °C showed an increasing acidification over time, except for the 0.1–0.4 mg⋅mL^−1^ PS AB formulations (containing 25 mM acetate buffer), which were only slightly more acidic after 48 weeks. As mentioned above, the acidic shifts in the unbuffered UW formulations were stronger than in the AB formulations, independent of PS concentration and PS type. From 0.1–10 mg⋅mL^−1^ PS, lower pH values were measured with increasing concentration, and the opposite trend is observed for concentrations ranging between 10–438 mg⋅mL^−1^ PS. The pH shifts in 0.2–1.0 mg⋅mL^−1^ PS20 UW formulations showed a delay of 2–4 weeks in comparison to the PS80 UW formulations. This is consistent with the content analysis via FMA and the FOX analysis, as both revealed a faster degradation of PS80 (Figure 2). The pH of 10–438 mg⋅mL^−1^ PS UW samples dropped immediately after storage at 40 °C to pH values of 2.9–5.2. The final pH value of the 10 mg⋅mL^−1^ UW formulations was about 0.3 pH lower for PS20 compared to PS80, and the difference grew to 0.8 pH units at concentrations of 100 mg⋅mL^−1^ PS. As mentioned, no or less severe acidic shifts were detected for the AB samples (Figure 3A, red and green lines). Within 18 weeks at 40 °C, the pH values of AB samples with PS concentrations of 1–438 mg⋅mL^−1^ decreased by 0.25–0.8 pH values. At these PS concentrations, the pH of PS20 AB samples showed a stronger acidic shift compared to PS80, while there were no discernible differences between the PSs at 0.1–0.4 mg⋅mL^−1^. For 438 mg⋅mL^−1^ of PS20 and PS80 in acetate, pH values outside the buffered range were measured, most likely due to the high volume proportion of PS material in the formulations. Nevertheless, the pH value dropped inside the buffered range after 10 weeks. The pH of all other AB samples stayed inside the buffered range over the entire study time.

Figure 3B shows the formic acid concentrations in the PS formulations after 48 weeks of storage at 40 °C. To facilitate a relative comparison of formic acid formation between the different PS concentrations, the formic acid concentrations were normalized to the PS content of 0.1 mg⋅mL^−1^. 438 mg⋅mL^−1^ and PS RM samples were not analyzed as the formic acid content per 0.1 mg⋅mL^−1^ PS was already negligible in 100 mg⋅mL^−1^ formulations compared to lower PS concentrations. Overall, the formic acid content per 0.1 mg⋅mL^−1^ PS decreased with increasing initial PS concentration between 1–100 mg⋅mL^−1^ PS. The highest content of formic acid was detected for 0.4 mg⋅mL^−1^. An unexpected behavior was observed for the formic acid measurements. Formic acid levels increased during the first 24 weeks, before they dropped and stayed constant for the 36 and 48-week samples (Figure 3C). As formic acid is one of the final oxidative degradation products, the concentration should be constant if there is no further degradation. However, a decrease after the maximum at 24 weeks was observed and evaporation directly before analysis, decomposition in water and carbon monoxide, or more likely incorrect determination at the 24 weeks timepoint could be an explanation.

For samples stored at 25 °C/60% rh, the same trends as for 40 °C storage were observed, however, the acidic shifts are weaker with slower kinetics (Figure 3C). Additionally, most of the UW formulations were still in the process of increased acidification after 48 weeks. The formic acid content increase at 25 °C was negligible (Figure 3D).

Samples stored at 5 °C showed only minor pH changes after 48 weeks as well as no increases in formic acid (Appendix A). The pH values of PS AB formulations stayed largely unchanged at pH 5.5–5.6 or 6.2 for the 438 mg⋅mL^−1^ PS20 formulation. In the first 4–12 weeks, a slight basification of 0.1–0.7 pH of the 0.1–1.0 mg⋅mL^−1^ UW samples could be observed.

### 3.3. Mass Spectrometry Analysis of PS20 and PS80

The PS20 and PS80 concentrations formulated in UW and AB for long-term storage were additionally analyzed by LC-MS to confirm the FMA results. Furthermore, information on specific degradation patterns as well as on the ratio of esterified to non-esterified PS species were obtained [11]. MS-based content analysis is reported to be more accurate, as FMA measurements detect only micelle-forming species in PS [58,61]. Figure 4A shows the total PS content normalized to the initial concentration for all PS20 and PS80 samples formulated in UW and AB stored at 40 °C/75% rh for 48 weeks. A comparable pattern between LC-MS and FMA results can be observed, however, with some differences. For the LC-MS results, higher initial PS concentrations formulated in UW and AB experienced less oxidative degradation. The final content after 48 weeks in all formulations is about 17 to 48% higher in comparison to the content measured via FMA (with greater deviations for higher concentrations) and there is no change in PS concentration for all 438 mg⋅mL^−1^ and RM formulations (Figure 4A). MS detection allows to quantify PS concentrations between 0.05 and 0.6 mg·mL^−1^ and is usually performed at a standard concentration of 0.2 mg·mL^−1^. Consequently, the highly viscous samples of 438 mg·mL^−1^ necessitate a dilution of more than 2000-fold, leading to a significant point-to-point variation. Given that all samples of a single concentration were prepared from a single master mix, any increases in concentrations during long-term storage can only be attributed to dilution inaccuracies or method variations. Additionally, the degradation kinetics for all PS types and formulations were slower. Comparing PS20 and PS80, it could be observed that PS80 formulations—independent of whether they were UW or AB—revealed stronger oxidative degradation, and PS20 AB formulations seemed to be more resilient against oxidative degradation (Figure 4A, red line). Using FMA measurements at 40 °C/75% rh, the differences between PS20 UW and PS20 AB were smaller (Figure 2A) compared to the MS content determinations (Figure 4A). In consensus with the FMA results, a delayed oxidation of the PS20 UW formulations was detected (Figure 2A and Figure 4A). Additionally, the unexpected drop of the 100 mg⋅mL^−1^ PS20 UW formulation detected by FMA could not be confirmed by LC-MS (Figure 2A and Figure 4A). A reason for the overestimated drop in the FMA content determination could be an increased polyester loss, resulting in an overestimation of the total content loss.

In MS-based detection approaches, PS species can be detected and selected based on their mass-to-charge values. Therefore, changes in individual species or in the esterified or non-esterified region of the peak pattern can be detected. In Figure 4B,C the peak patterns of 10 mg⋅mL^−1^ PS20 and PS80 in AB stored at 40 °C/75% rh are shown. The concentration of 10 mg⋅mL^−1^ PS20 and PS80 was chosen to avoid fast oxidation processes and to enable easier monitoring of the differences. Generally, only a small increase in the non-esterified peak between 8 and 12 min was recorded for both polysorbates. However, drastic decreases in the higher-order ester fractions for PS20 (retention times > 20 min) and PS80 (retention times > 22 min) are observed (Figure 4B,C). This provides some hints for the faster degradation of higher-order species, however, the conversion of multi-esters into monoesters must be considered as well and a more detailed analysis would be necessary for an unequivocal verification of this observation. This issue is later discussed in more detail (see next chapter). For PS20, only marginal decreases of the PS species esterified to C8 (retention times: 13 to 14 min) and C10 (retention times: 15 to 16 min) can be identified (Figure 4B). Moreover, PS20 esterified to lauric acid (C12; retention time ~17 min) seems to be more resilient against oxidation; however, the degradation of lauric acid polyesters into lauric acid monoesters must be considered as well. For PS80 oxidation, new species emerge between 15 and 18 min, together with a baseline shift to higher intensities, especially for early retention times (10–15 min) (Figure 4C). 

As mentioned, MS-based detection allows for the selective quantification of individual species in the heterogeneous PS mixtures by filtering for specific mass-to-charge values and retention times. Thereby, the resilience of PS species esterified to middle-chain length fatty acids such as caprylic acid (C8) and capric acid (C10), which was observed in the peak pattern of PS20 AB 10 mg·mL^−1^ at 40 °C, can be analyzed in detail. The intensities of all POE monoester variants for caprylic acid (C8), capric acid (C10), lauric acid (C12), myristic acid (C14), palmitic acid (C16), stearic acid (C18), and oleic acid (C18:1) were extracted and normalized to the initial value (*t* = 0). Polyesters of the corresponding fatty acids were not considered, as their content is neglectable for most of the fatty acids. The POE monoester variants are, for instance, sorbitan-C12 species with 18 to 33 numbers of POE units in total. Their changes over time are illustrated in Figure 4D. As assumed from the peak patterns of PS20 AB, differences in the oxidative susceptibility of different esterified fatty acids are observed (Figure 4B). In particular, the PS species esterified to middle-chain length fatty acids (C8 and C10) reveal only minimal degradation, followed by PS molecules esterified to lauric acid with slightly increased degradation (Figure 4D). Finally, polysorbate variants with longer fatty acids such as myristic, palmitic, and stearic acid degraded more severely, and the fastest oxidation was observed for oleic acid (Figure 4D). The same analysis was conducted for 10 mg·mL^−1^ PS80 AB and the results are depicted in Appendix A. Here, PS80 monoesters, esterified to linoleic acid (C18:2), revealed the fastest degradation with less than 10% left after 2 weeks, directly followed by oleic acid (C18:1) with approximately 40% left after 4 weeks (Appendix A). For the saturated fatty acids such as palmitic or stearic acid, no decrease in content was observed (Appendix A). For this analysis, only fatty acids with higher than 1.0% content were considered, as the intensities are very low for these low abundant fatty acid species.

### 3.4. Oxidation Markers of PS80

Different oxidation markers for PS80 were analyzed based on their mass-to-charge values and retention times. The intensities normalized to the initial concentration of sorbitan mono-oleate ethoxylated with 26 and 20 POE moieties (C18:1 26 POE and C18:1 20 POE), sorbitan di-oleate ethoxylated with 26 POE units (2x C18:1), as well as oxidation products of sorbitan di-oleate (26 POE moieties), with one of both oleic acids oxidized to hydroxyl-C18:1 (C18:1 + C18:1-OH) or hydroperoxyl-C18:1 (C18:1 + C18:1-OOH) were extracted and plotted over time (Figure 5). The 10 mg·mL^−1^ PS80 AB formulations were used, as the rapid oxidation processes were occurring at lower PS80 concentrations. Oxidation in the vicinity of the double bond was observed as species such as C18:1 + C18:1-OH and C18:1 + C18:1-OOH emerged due to favorable H-atom abstraction in the vicinity of the double bond, with a maximum after 4 weeks of storage at 40 °C/75% rh (Figure 5). Additionally, sorbitan di-oleate ethoxylated with 26 POE units (2× C18:1) was investigated as well. In general, a faster degradation of the di-oleate species was observed in comparison to mono-oleic acid esters ethoxylated to 26 and 20 POE moieties. C18:1 20 POE species were used as control, as cleavage within the POE chain of 2× C18:1 would result in C18:1 species with less POE moieties in general. Only a small decrease in the C18:1 + C18:1-OH content was observed after 4 weeks.

## 4. Discussion

### 4.1. Oxidation as the Root Cause for PS20 and PS80 Degradation

Several methods were used to confirm that oxidation was the root cause for the PS degradation in the long-term stability study. First, sample conditions were selected where neither enzyme-mediated hydrolysis nor chemical hydrolysis of PS was expected to occur. Therefore, PS samples were stored in the absence of proteins/antibodies and formulated at pH values of approximately 5.5 and 6.5 [66]. Additionally, an increase in ROS as well as a continuous acidification were monitored under accelerated storage conditions via FOX assays and pH measurements. These parameters are reported to be associated with oxidative PS degradation [7,9]. Furthermore, oxidation markers of PS80 were analyzed, clearly revealing oxidative degradation. The presence of metal ions and/or peroxide impurities in the raw products are discussed to be related to oxidative degradation [9,25,26,27,28,29,30,31,32,33]. Therefore, ICP-MS measurements were performed to evaluate possible impurities in the used raw materials. PS20 and PS80 RMs as well as 1 M acetate buffer were analyzed for relevant metal content before the study was initiated. Iron, molybdenum, and boron were found as impurities above the detection limit (Table 1). In particular, the presence of iron (141.8 ng⋅mL^−1^ and 102.2 ng⋅mL^−1^ in RMs of PS20 and PS80, respectively) in combination with tiny amounts of H_2_O_2_ or other organic peroxides can catalyze the generation of ROS via Fenton/Fenton-like reactions [7,37], Haber–Weiss reactions [34,35,36], or other redox reactions with organic substrates, peroxides, and molecular oxygen [27,39,47]. The presence of 2330.5 ng⋅mL^−1^ boron and 12.7 ng⋅mL^−1^ molybdenum in the 1 M acetate buffer is not expected to initiate oxidative processes (Table 1). Furthermore, samples of 100 mg⋅mL^−1^ PS20 and PS80 UW were analyzed after storage for 48 weeks at 40 °C to evaluate potential leachables over the time course of the study (Table 1). Here, iron (Fe) and aluminum (Al) concentrations of 44.1 ng⋅mL^−1^ and 1233.5 ng⋅mL^−1^ were detected for PS20 UW as well as 23.9 ng⋅mL^−1^ and 572.2 ng⋅mL^−1^ for PS80 UW, respectively. These iron concentrations can be considered as an increase in respect to the initially used raw material, since for 100 mg⋅mL^−1^ PS80 or PS20 a dilution factor of approximately 10 or 11 can be calculated, respectively. Thereby, an iron concentration of 23.9 and 44.1 ng⋅mL^−1^ can be extrapolated to ca. 254 and 489 ng⋅mL^−1^ iron in PS80 and PS20 raw material after storage for 48 weeks at 40 °C, respectively. The raw material stored for 48 weeks at 40 °C could not be analyzed due to a lack of material. Additionally, small concentrations of copper as well as an increase in the boron concentration were monitored. In particular, the increases in aluminum and boron were expected as potential leachables in the glass vials used in this study [67,68], however, the dissolution of glass elements is dependent on the formulation conditions that were used as well as on the glass type [69,70]. The presence of aluminum ions was reported to induce fatty acid particles in the case free fatty acids are generated e.g. by enzymatic degradation [71], and an impact on oxidation can be discussed. It was also reported to have a propensity to be involved in the oxidation of oils [72]. Independent of the leachables, increased temperatures can accelerate the oxidation of polysorbates [9], as activation energies as well as reaction kinetics can be affected by temperature.

### 4.2. Concentration-Dependent Oxidative Degradation of PS20 and PS80

At 25 and 40 °C, PS concentration-dependent oxidation was observed, independent of PS type and formulation (Figure 2A,C). No degradation was monitored for the PS20 UW and AB formulations at 5 °C for up to 48 weeks, whereas PS80 formulations revealed oxidative degradation starting between 36 and 48 weeks (Appendix A). Tracking the concentration of ROS species at 25 and 40 °C, which can be intermediates in the radical chain reaction process of oxidation, revealed that the oxidative processes in the high-concentration PS regime (10–100 mg·mL^−1^ PS) did indeed slow down in comparison to the low-concentration PS formulations (0.1–1.0 mg·mL^−1^), even though significantly higher proportions of the PS material remained. Possible explanations for the surfactant concentration-dependent degradation of PS20 and PS80 could be (i) the changing environment in the concentrated solutions and (ii) the depletion of oxygen or the presence of specific ROSs. Mittag et al. (2022) showed via EPR and spin trapping experiments that the radical formation from PS80 raw material is different to aqueous dilutions of PS80 (10% (*w*/*v*)) [49]. In bulk PS80 solutions, mainly ROO^•^ and RO^•^ species, were detected, whereas in 10% (*w*/*v*) PS80 solutions mainly HO^•^ radicals were measured [49]. As HO^•^ radicals react in a diffusion-controlled manner (*k* > 10^9^ M^−1^·s^−1^) [50,51,52], higher degradation could be assumed. In contrast, ROO^•^ and superoxide radicals possess reaction constants six to nine orders of magnitude lower and might therefore decelerate the oxidative degradation in higher concentration PS formulations [49,50,51,52,53]. Additionally, other solution properties, such as viscosity or oxygen solubility, could have an impact as well. For instance, viscosity is affected by the actual concentration of polysorbates, potentially decelerating oxidation for high concentrations of polysorbates, as HO^•^ radicals react in a diffusion-controlled manner. The second way in which higher concentrations of polysorbates could affect oxidation is the availability of oxygen. Molecular oxygen is essential for oxidation reactions, as it is inserted early in the oxidative chain reaction [9,57]. By limiting the availability of oxygen, the oxidative degradation processes of polysorbate can be suppressed [23]. We analyzed the headspace oxygen in the vials after storage for 48 weeks at 40 °C/75% rh, 25 °C/60% rh, and 5 °C (Appendix A). For 40 °C, decreasing amounts of oxygen of approximately 3% (*v*/*v*) were detected with increasing PS concentrations (10 and 100 mg·mL^−1^), whereas for the lower PS concentrations (0.1–1 mg·mL^−1^), 8 to 12% (*v*/*v*) of remaining oxygen was detected in the headspace (Appendix A). The oxygen levels were comparable for the different PS types and formulations, reaching levels of approximately 0 to 3% oxygen in the headspace for 100 mg·mL^−1^ of PS for 25 and 40 °C (Appendix A). This observation seems plausible, as the ratio between available oxygen and PS molecules diminishes drastically with increasing polysorbate concentration. With some assumptions the molar ratio between O_2_ and polysorbate can be estimated (see Appendix A). For concentrations of 10 and 100 mg·mL^−1^ PS, the molar ratios between O_2_ and polysorbate molecules are approximately 1:0.8 and 1:8, respectively. Therefore, drastically less oxygen is available per PS molecule, which could slow down or entirely stop the oxidation process. The decreasing H_2_O_2_ equivalent concentration per 0.1 mg·mL^−1^ PS with rising initial PS concentration is in line with less oxidative degradation with increasing initial PS concentration (compare Figure 2). The FMA results were verified by MS-based PS content analysis. The used storage conditions of polysorbate are comparable to diluent or placebo formulations in the absence of proteins, which can be used to dilute the active pharmaceutical ingredient before application. As mentioned, reduced headspace oxygen content is considered a major factor in slowing down the oxidation processes at high polysorbate concentrations. These observations are in line with general polysorbate raw material storage conditions, in which small (preferential light-protected) containers are used and are covered with an inert gas to limit the oxygen accessibility [22,60]. The presence of proteins would affect the oxidation of polysorbate as well, as elaborately discussed and summarized in Weber et al. (2023) [60]. Radicals formed upon PS oxidation can additionally react with proteins, causing their oxidation. As proteins are normally present in high concentrations, reduced oxidative polysorbate degradation can be observed [27,30,54]. The protein oxidation is not intended as it could influence the drug efficacy. Based on such results, mid- to long-term storage of polysorbate-containing solutions at 5 °C with limited access to oxygen by either covering the headspace in the vials with an inert gas or potentially by filling the vials at higher volumes could serve as mitigation approach. Nevertheless, it is worth mentioning that performing appropriate handling studies allow to guide proper mitigation.

The bigger difference between PS oxidation at 25 °C in comparison to 40 °C can be explained by the temperature itself. As mentioned above, temperature has different effects on reactions, namely influencing activation energy and/or doubling the chemical reaction rates by increasing the temperature by 10 K.

pH changes induced by oxidation are dependent on the initial concentration of poly-sorbate, as higher acidifications are observed with increasing PS concentrations, peaking at 10 mg·mL^−1^ (Figure 3A). Oxidation of increased PS concentrations generates higher absolute numbers of short-chain acids, leading to higher acidifications. The actual pH value can have a considerable impact on the oxidation process, as for instance, Fenton or Fenton-like reactions, which can initiate the oxidative chain reaction, possess higher reaction constants at lower pH values [44,45,46]. However, other pH-dependent redox reactions could also have an influence. The delayed oxidation of PS20 UW in the concentration range between 0.1 mg·mL^−1^ and 1.0 mg·mL^−1^ can be explained by the same phenomenon but will be discussed in detail in the next chapter. In general, it was shown that PS oxidation (PS20 and PS80) is dependent on the initial PS concentration, with less oxidation occurring at higher PS concentrations. The formation of different radicals as well as the availability of oxygen were discussed as potential root causes.

### 4.3. PS20 vs. PS80 Oxidation

FMA, ROS, and MS-based content determinations of PS revealed differences between the oxidation of PS20 and PS80. For FMA at 40 °C, smaller differences between PS20 and PS80 oxidation were observed for concentrations between 0.1 mg·mL^−1^ and 10 mg·mL^−1^, whereas MS-based approaches revealed a stronger difference between both polysorbates, with PS80 being more susceptible to oxidation. Additionally, faster and more severe oxidation of PS20 and PS80 was observed using FMA content determinations in comparison to the MS-based method (Figure 2A and Figure 4A). Differences related to the PS quantification between indirect content determination by FMA and direct measurements by CAD or MS are reported in the literature [58]. The current working hypotheses for the observed differences in PS content are linked to the physico-chemical properties of specific PS fractions (monoesters and polyesters) and especially their tendency to form micelles, in which the FMA dye is incorporated. The FMA is sensitive to the partitioning of the fluorescence dye NPN within micelles. Lippold et al. (2017) have discussed this in detail [58]. They concluded that “it was particularly surprising to find that the main component of PS20, polyoxyethylene sorbitan monolaurate, did not show a signal at the studied concentration using FMA. Moreover, the degradation of polysorbate poly-esters, was reflected much stronger in FMA than MM-CAD results” [58]. Additionally, Tomlinson and colleagues (2020) characterized different fractions of polysorbate and reported a roughly two orders of magnitude higher critical micelle concentration (CMC) for monoester fractions in comparison to the diester fractions of PS20, as well as a three-fold difference between all PS80 fractions [73]. In their study, the authors did not consider PS triester fractions in their analysis. These differences impact the results of the FMA drastically, especially for the preferred degradation of certain species, which is usually observed for the oxidation of PS. Additionally, PS20 and PS80 diesters have higher aspect ratios with more micellar association compared to their corresponding monoester fractions [74], which will impact the FMA analysis as well. Preferential degradation of higher-order esters, which is observed in the peak pattern (Figure 4B,C), results in higher degradation detected in the FMA content analysis, as the species with the lowest CMC and highest hydrophobicity are preferentially degraded [58]. Additionally, the total PS content is shown in Figure 4A, which is composed of the esterified (PS20 retention times: 12–30 min; PS80 retention times: 12–27 min) as well as the non-esterified species (PS20/80 retention times: 7–12 min). The non-esterified molecules (POE and sugar derivates) are normally not expected to be involved in micelle formation as they are not amphiphilic. However, fatty acids such as lauric acid or oleic acids can be part of a micelle structure. For instance, strong degradation of the esterified PS is observed after 48 weeks at 40 °C for 10 mg·mL^−1^ PS80 AB (Figure 4C), however, approximately 60% of the total amount is still measured (Figure 4A). Therefore, a direct comparison between MS and FMA results must be performed cautiously. The different oxidative degradation susceptibilities are also the root cause for the different trends of degradation observed for the 438 mg·mL^−1^ concentration in FMA and MS content determination (compare Figure 2A and Figure 4A). In the case of FMA, a content reduction is likely due to a minor decrease in higher-order esters, which significantly influence the FMA content determinations. Nevertheless, rapid oxidative degradation was observed for PS80 formulations using MS detection as well as FMA analysis, especially when comparing PS20 and PS80 in their corresponding formulations. The faster oxidation of PS80 can be attributed to the higher oxidizability of PS80, based on the higher amount of unsaturation sites [57]. Hydrogen abstraction in the vicinity of the double bond is energetically favored in comparison to hydrogen abstraction at the POE region [57,75]. To initiate the radical chain reaction of oxidation, lower hydrogen atom abstraction energies could be beneficial. The higher oxidative susceptibility of PS80 has often been reported in the literature [7,24,31,54,55,56,57], however, an in-depth comparison in a concentration-dependent manner is missing in the literature, as additional stressors such as iron, H_2_O_2_, light, or 2,2′-azobis(2-methylpropionamidine) dihydrochloride (AAPH) have normally been used to accelerate the oxidation process. For instance, AAPH is a water-soluble radical starter [59] which is normally not found in pharmaceutical products. The differences between PS20 and PS80 oxidation could be reduced for higher concentrations such as 438 mg·mL^−1^, when, in general, less oxidation was observed, as discussed above. At lower temperatures, greater differences between PS20 and PS80 were observed. At 5 °C, no oxidative degradation was measured for PS20 formulations in UW and AB pH 5.5 (independent of the PS concentration), whereas most PS80 formulations revealed oxidation between 36 and 48 weeks (Appendix A). As extensively discussed, the absolute values in content reduction for PS80 must be considered with caution, as the FMA assay is highly affected by higher-order species, which are the most susceptible species for oxidative degradation. With reduced temperatures, less oxidative stress is applied, potentially affecting the likelihood of radical formation. As the H-abstraction and radical initiation in oleic acid is generally favored [57], more initiation reactions should occur in PS80, propagating the oxidation reaction and thereby increasing the difference between PS20 and PS80.

PS20 UW formulations in the lower concentration regime (0.1–1.0 mg·mL^−1^) revealed a degradation delay at 40 °C, which was even increased at 25 °C. As no additional impurities such as iron or other divalent ions were found in the acetate formulations (see ICP-MS of 1 M acetate buffer), the pH difference between UW (pH > 6.0) and AB (pH 5.5) could be a potential root cause, as the Fenton reaction is more likely to occur in acidic conditions [44,45,46]. For instance, Xu and colleagues (2009) investigated the pH dependence of the Fenton process via the degradation of melatonin, revealing faster kinetics at acidic pH values [45]. This pH dependence could be used to explain the lag phase of oxidation for PS20 UW formulations, as it could be argued that with higher pH values in water formulations (pH > 6.0) less oxidation initiations via the Fenton reaction could occur. However, an impact of other redox reactions with metals could also be assumed. The importance of the pH value is also observed at later timepoints for the PS20 formulations (Figure 4A). Although oxidation of the PS20 UW formulations exhibited an initial lag phase, stronger degradations (lower total PS20 concentrations) are observed after 48 weeks (Figure 4A, black and red line). The pH dependence of the Fenton reaction or another unknown redox reaction could explain these observations as well. As the UW formulations are not buffered, the pH values drastically decrease after approximately 12 weeks into strongly acidic ranges with pH values of approximately 3 (Figure 3A). At these pH values of approximately 3, the Fenton reactions are far more likely than at pH values of 5 [44,45,46]. Thereby, more oxidation processes could potentially be triggered in PS20 UW concentrations, resulting in more oxidized/degraded PS20 after 48 weeks. To illustrate the correlation of pH and PS degradation an overlay of both is shown in Appendix A. The observed higher ROS amounts in the UW formulations would also fit to this explanation (Figure 2). For PS80 UW formulations, no lag phase is observed at 25 and 40 °C, however, MS content analysis also revealed an increased degradation for both UW formulations after 48 weeks (Figure 4A). For 5 °C, a difference between PS80 UW and the AB formulation was monitored, supporting the general idea of pH-dependent reactions. Nevertheless, other factors, such as iron solubility at different pH values or the potential impact of acetate or the counter ion (sodium), could also affect the oxidation of PS between the UW and AB formulations. A working hypothesis explaining the absence of a lag phase for PS80 UW oxidation at high concentrations could be that there is a generally higher potential for oxidation initiation. As oxidation initiation is already energetically favored in the vicinity of the double bond, the difference caused by the different pH values is weakened at higher temperatures. To validate this hypothesis related to pH dependent oxidation, more experiments are necessary, which are outside the scope of this study.

One striking difference between the chromatograms of PS20 and PS80 is the baseline shift in PS80 formulations (Figure 4C). This phenomenon was observed before and has been attributed to new, more hydrophilic species such as keto-oleic acid, hydroxyl-oleic acid, hydroperoxyl-oleic acid, or cleavage products of oleic acid attributed to the oxidation in the vicinity of the double bond [25,26,31,55,59].

### 4.4. Oxidation Marker and Fatty Acid Dependent Oxidation of Polysorbates

The MS-detection allows us to compare the oxidative degradation of individual species in polysorbates by selecting specific mass-to-charge values and retention times. Here, we observed the different oxidation properties of polysorbates, dependent on the length of the esterified fatty acid chain. Polysorbate monoesters esterified with shorter fatty acids are more resilient against oxidation (Figure 4D). This phenomenon was also reported by Borisov et al. (2015) and Zhang et al. (2018), with both revealing PS species esterified to longer fatty acids as more prone to oxidation while using the artificial oxidation accelerator AAPH [55,59]. The exact rationale behind this observation is unknown, however, preferential ether bond scissions within the esterified POE chain were discussed based on different arrangements of the POE chain [55]. It is assumed by the referred studies that the POE arms esterified to long fatty acids possess less degrees of freedom and therefore adopt an all-trans or physically stretched conformation with disturbed H-bonding networks [76]. Usually, the trans-gauche-trans helical formation of POE chains is stabilized by trapped water molecules, which help to provide an extensive hydrogen bond network [77,78]. The physically stretched conformation, in combination with the disturbed H-bonding network and the resulting destabilized water molecules are assumed to be the origin for facilitated oxidation of POE arms esterified to longer fatty acids [55]. However, this working hypothesis has yet to be proven. Additionally, a faster oxidation of C18:2 was observed in comparison to C18:1, illustrating that the higher degree of unsaturation results in faster oxidation. This accelerated oxidation of C18:2 was observed before by Hvattum and colleagues (2012), revealing the complete degradation of C18:2 polysorbate species after 8 weeks of storage at 40 °C under air [56]. This higher susceptibility towards oxidation can be explained by the even more favorable H-atom abstraction energy between two double bonds in comparison to the H-atom abstraction energy in the vicinity of one double bond [57]. For instance, at 100 °C, linoleic acids have 10-times higher relative rates of oxidation in comparison to oleic acid [72].

Additionally, the normalized intensities of the oxidation products of PS80 AB (10 mg·mL^−1^) were analyzed, revealing oxidation in the vicinity of the double bond of oleic acid, as hydroxyl-C18:1 or hydroperoxyl-C18:1 species are formed from diesters of PS80 (Figure 5). These species are intermediates of oleic acid oxidation and are subsequently further degraded, as observed by the drop after 4 weeks. Degradation into smaller fatty acids such as 1,9-nonanedioic acid [54] or 2-decenedioic acid have already been shown [31]. MS-based approaches revealed the highest intensities for sorbitan esters ethoxylated with 26 POE units. To evaluate the oxidation of higher-order esters in comparison to monoesters in PS80, the normalized intensity of sorbitan mono-oleic acid esters with 26 POE moieties was compared to sorbitan di-oleic acid esters with 26 POE units. The POE chain oxidation processes of sorbitan di-oleate in the POE chain would be expected to result in PS80 mono-oleic acid ester variants with shorter POE numbers. As diesters degrade faster, a preferred oxidation of diesters can be concluded. Additionally, a slightly delayed oxidation of PS80 mono-oleate ethoxylated with 20 POE units was monitored. These delays result from the cleavage of oleic acid esterified to 6 POE units from sorbitan di-oleic acid esters with 26 POE moieties.

In general, polysorbate oxidation in formulations is relevant, as a reduction in the polysorbate concentration may lead to inadequate protection of the active pharmaceutical ingredient and the formation of problematic degradation products. The precise concentration of PS, which can degrade without causing stability issues, depends on the protein itself. To the best of the authors’ knowledge, no reports have been published thus far on the general harm of specific oxidation degradation products on protein molecules.

In the present case study, we focused on the analysis of PS20 and PS80 HP from a single supplier, as our primary focus was on directly comparing PS20 and PS80, as well as their varying concentrations. The impact of different suppliers and other qualities was outside the scope of this particular study. However, we anticipate similar results when comparing PS20 and PS80 from different vendors, as we could show that the presence of oleic as well as linoleic acid are the main contributing factors for oxidation, which should not drastically change between vendors as long as the peroxide and metal content are similar. For the comparison of polysorbates of HP, super refined (SR), or even purer qualities (e.g., all-oleate), no definitive statement can be made. For a description of the current literature, see Weber et al. (2023) [60], summarizing from the current literature that PS20 all-laurate and PS80 all-oleate possess higher oxidative susceptibility in comparison to their multicompendial counterparts [25,26,27,31].

## 5. Conclusions

The oxidative degradation susceptibility of different PS20 and PS80 concentrations ranging from 0.1 mg·mL^−1^ to raw materials formulated in water and acetate buffer pH 5.5 was investigated at 5, 25, and 40 °C to achieve an in-depth comparison between PS20 and PS80, as well as to elucidate the concentration dependence of PS oxidation. Especially at 5 °C, a concentration-dependent comparison was missing in the literature, as oxidation accelerators were mainly used to compare both PS types. We showed that PS20 and PS80 oxidation is dependent on the initial concentration, with decreasing oxidative susceptibility at increasing initial PS concentrations, independent of PS type and whether they were formulated in water or acetate. Especially for concentrations above 438 mg·mL^−1^, only minor degradation was observed by MS detection. The availability of oxygen as well as the generation of different radicals were discussed as root causes. Mittag and colleagues (2022) reported ROO^•^ and RO^•^ to be the predominant species for raw materials of PS80, whereas mainly HO^•^ radicals were measured in 10% (*w*/*v*) PS80 solutions, independent of the quality grade [49]. Additionally, we found drastically diminished oxygen concentrations in the headspace (approximately zero) for high concentrations of PS stored for 48 weeks at 25 and 40 °C, diminishing and terminating oxidation. In general, faster degradation resulting in less PS content was observed with increasing temperatures. For 5 °C, no degradation of PS20 was observed up to 48 weeks, whereas oxidative degradation of PS80 initiated between 36 and 48 weeks. Comparing PS20 and PS80, a higher degradation of PS80 was confirmed in this case study, which is most likely due to the higher amount of unsaturated fatty acids and their higher oxidizability when compared to their saturated counterparts [57]. The absolute values in the content reduction for PS80 were detected by FMA and must be considered with caution, as mainly higher-order esters degrade initially in oxidation processes, which is discussed in detail above. Especially at lower temperatures, pronounced differences between PS20 and PS80 are monitored, as for 5 °C an earlier onset of degradation is seen for PS80. Furthermore, PS20 UW formulations revealed a lag phase for the oxidative degradation at 25 and 40 °C, and differences in the pH value and the pH dependence of the Fenton reaction or other redox reactions were discussed as root causes. As Fenton reactions occur less frequently at higher pH values, especially for pH values higher than 6, the frequency of initiation reactions and the resulting PS degradation could be diminished for more physiological pH values [44,45,46]. Nevertheless, other redox reactions in the presence of iron could also be affected by the different pH values between water and acetate buffer formulations. In summary, the following phenomena were observed:-PS80 is more prone to oxidation than PS20 as it possesses an earlier onset of oxidation, independent of whether it is formulated in water or acetate buffer;-At 5 °C, PS20 showed no oxidative degradation in water and acetate buffer up to 48 weeks, whereas PS80 oxidation started after 36 weeks;-Lower temperatures (less oxidative stress) revealed larger differences between PS20 and PS80 oxidation;-Delayed oxidation was monitored for PS20 formulated in water and high pH values were discussed as a potential origin;-Oxidation is dependent on the initial PS concentration, most likely due to differences in radical generation and/or due to limiting oxygen availability;-Metal ions such as iron in the presence of oxygen are most likely the root cause of oxidation.

## Figures and Tables

**Figure 1 pharmaceutics-15-02332-f001:**
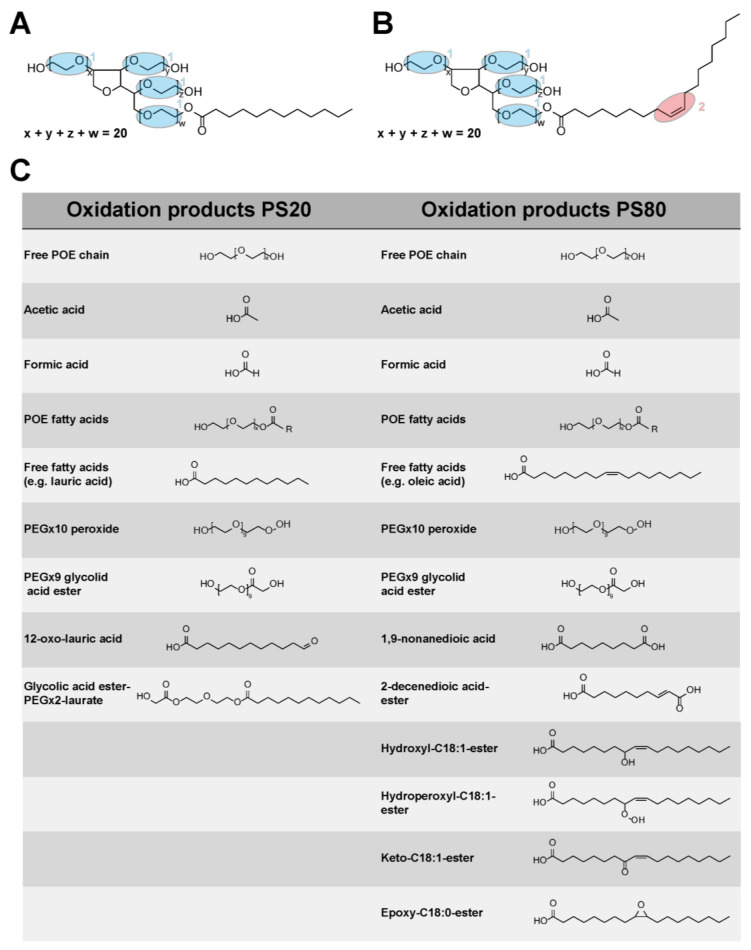
Structures and oxidative degradation products of PS20 and PS80. (**A**) General structure of PS20 with potential radical initiation sites on the POE units (1, blue). (**B**) General structure of PS80 with potential radical initiation sites on the POE units (1, blue) and near the unsaturated site of oleic acid (2, red). According to the pharmacopoeias, x + y + z + w = 20. (**C**) Selected degradation products of polysorbate due to oxidation. Non-esterified products are illustrated. The double bond increases the variety of possible degradants of a PS molecule. Some of the molecules are not drawn to scale or with the corresponding isomer with respect to the double bond to save space.

**Figure 2 pharmaceutics-15-02332-f002:**
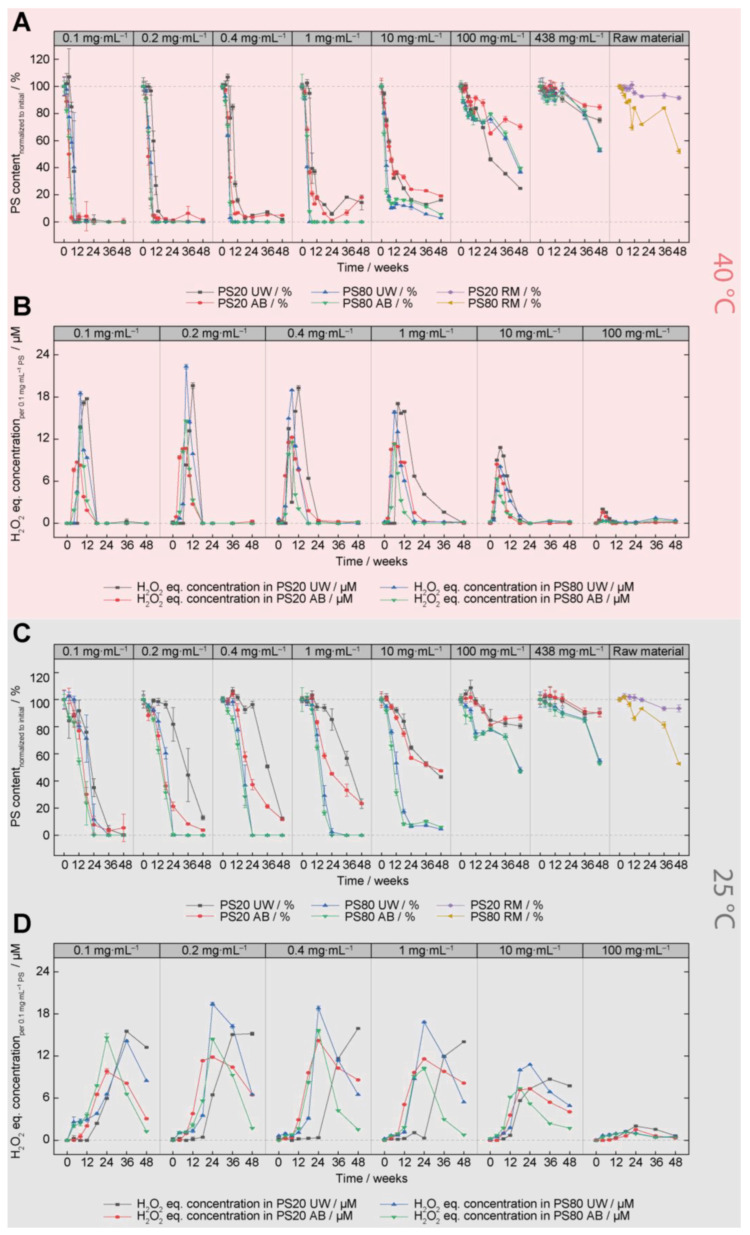
Impact of polysorbate concentration, temperature, and storage time on polysorbate content (**A**,**C**) and peroxide formation (**B**,**D**). (**A**,**C**) PS20 and PS80 contents were determined for different initial PS concentrations in water (UW) and 25 mM acetate buffer (AB) pH 5.5 stored for 48 weeks at 40 °C/75% rh (**A**) and 25 °C/60% rh (**C**) via FMA. The data were normalized to the initially measured PS concentration of each formulation. Two biological replicates of each sample (*n* = 2, except for the raw materials) were measured with four technical replicates each (*n* = 4). The reference lines mark PS concentrations for 0 and 100% of the initial values. (**B**,**D**) H_2_O_2_ equivalent concentration for PS20 and PS80 concentrations of 0.1–100 mg⋅mL^−1^ PS formulated in UW and AB pH 5.5 for storage for 48 weeks at 40 °C/75% rh (**B**) and 25 °C/60% rh (**D**) measured via the FOX assay. The data are presented as H_2_O_2_ equivalent concentrations per 0.1 mg⋅mL^−1^ PS in the formulations. Three technical replicates of each sample were measured (*n* = 3). The reference line marks a H_2_O_2_ equivalent concentration of 0 µM.

**Figure 3 pharmaceutics-15-02332-f003:**
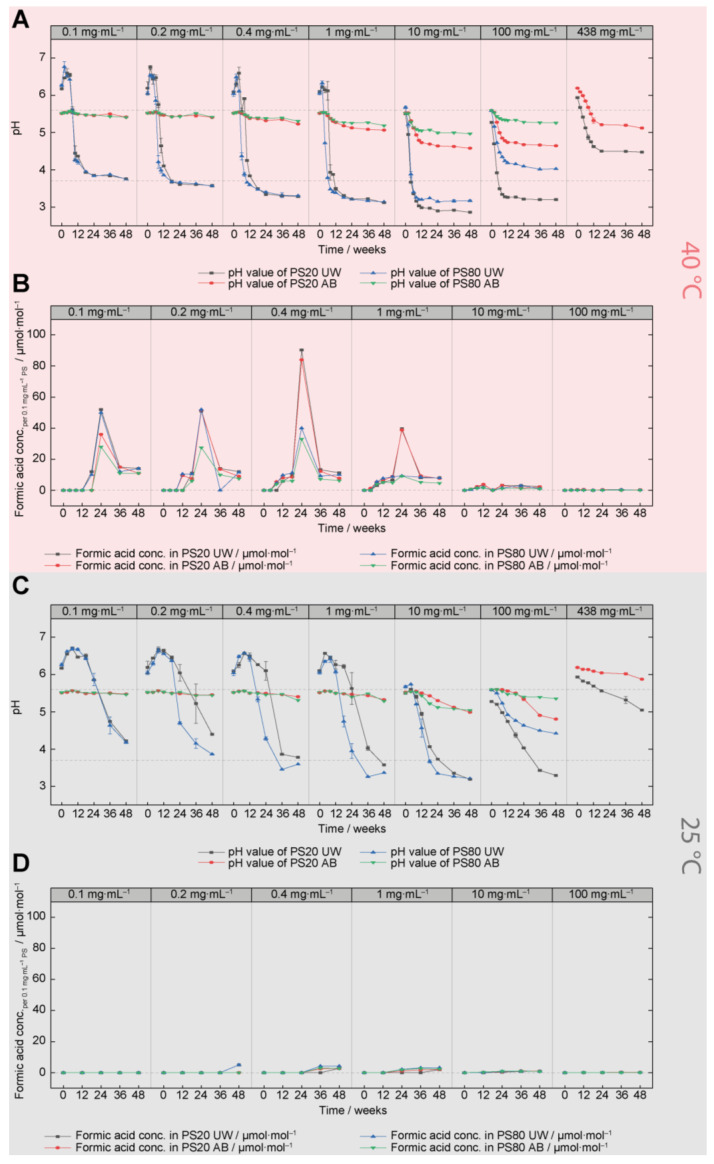
Impact of polysorbate concentration, temperature, and storage time on the pH value (**A**,**C**) and on formic acid formation (**B**,**D**). (**A**,**C**) pH values of all PS formulations formulated in water (UW) and 25 mM acetate buffer (AB) pH 5.5 used in the stability study after 48 weeks of storage at 40 °C/75% rh (**A**) and 25 °C/60% rh (**C**). Two biological replicates of each sample were measured (*n* = 2). The 438 mg⋅mL^−1^ PS80 formulations and PS raw material samples were not measured due to high viscosities. Reference lines mark the buffer range of the used 25 mM acetate buffer (pH 3.7–5.6). (**B**,**D**) Formic acid concentrations for PS formulations in UW and AB pH 5.5 for concentrations of 0.1–100 mg⋅mL^−1^ used in the stability study after 48 weeks of storage at 40 °C/75% rh (**B**) and 25 °C/60% rh (**D**) measured via GC. The data are presented as formic acid concentration per 0.1 mg⋅mL^−1^ PS in the formulations. One replicate of each sample was measured (*n* = 1). The 438 mg⋅mL^−1^ PS formulations and PS raw material samples were not measured. The reference line marks a formic acid concentration of 0 µmol⋅mol^−1^.

**Figure 4 pharmaceutics-15-02332-f004:**
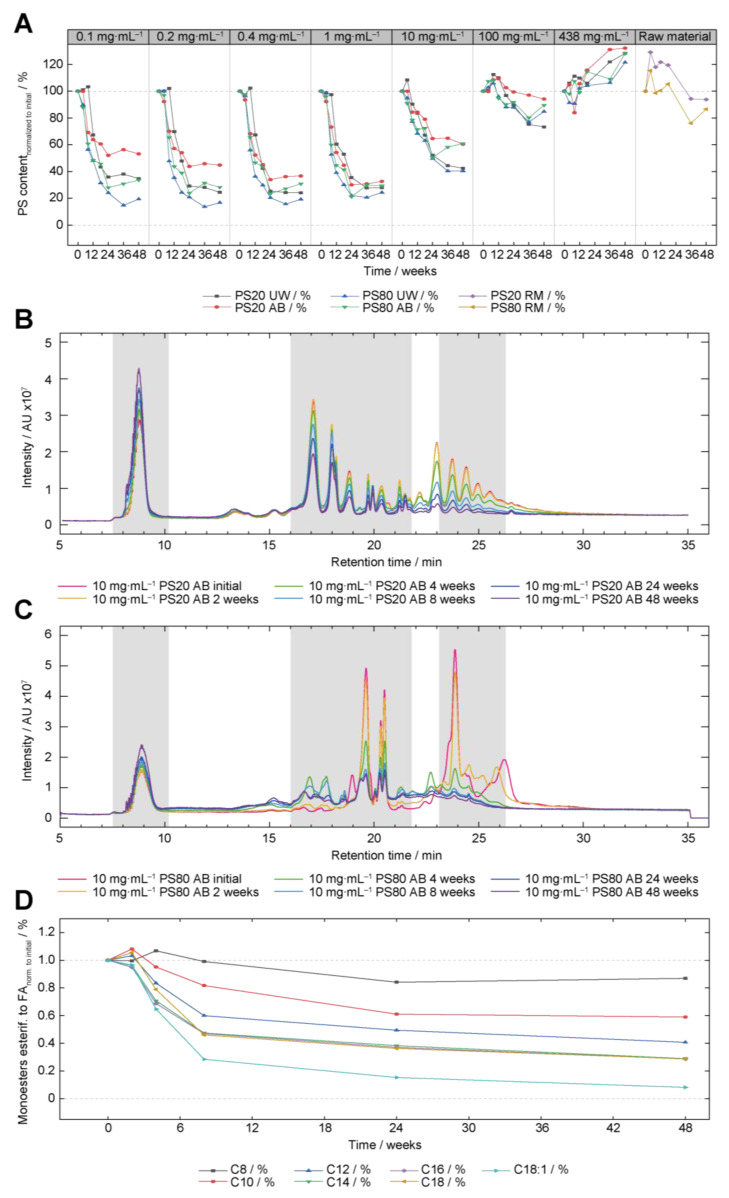
RP-UPLC-MS measurements of PS20 and PS80 after storage at 40 °C. (**A**) PS20 and PS80 content was determined for different initial PS concentrations in water (UW) and 25 mM acetate buffer (AB) pH 5.5 stored for 48 weeks at 40 °C/75% rh via RP-UPLC-MS. The data are presented as% of the initially measured PS concentration of each formulation. One replicate of each sample was measured (*n* = 1). The reference lines mark PS concentrations of 0 and 100% of the initial values. (**B**) Evolution of the peak pattern of 10 mg⋅mL^−1^ PS20 AB during storage at 40 °C. One replicate of each sample was measured (*n* = 1). The gray-colored areas mark the retention time windows from left to right of non-esterified (~7.5 to 10 min), monoester (~16 to 22 min ), and polyester (~23 to 26 min) species. (**C**) Evolution of the peak pattern of 10 mg⋅mL^−1^ PS80 AB during storage at 40 °C (measurements and layout equivalent to (**B**)). (**D**) Preferential oxidation of polysorbate monoester species esterified to different fatty acids. The peak patterns of 10 mg⋅mL^−1^ PS20 AB at 40 °C were used to extract the intensities of the different POE variants esterified to caprylic acid (C8), capric acid (C10), lauric acid (C12), myristic acid (C14), palmitic acid (C16), stearic acid (C18), and oleic acid (C18:1) over time. The intensities were normalized to the corresponding initial values (*t* = 0) and plotted versus time.

**Figure 5 pharmaceutics-15-02332-f005:**
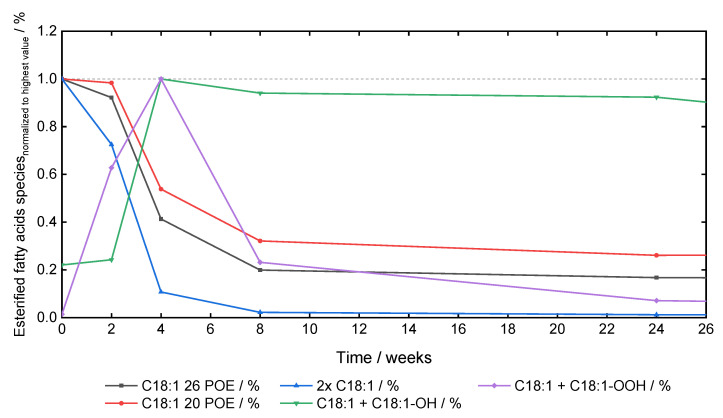
Normalized ester content of PS80 oxidation products after storage at 40 °C. The peak patterns of 10 mg⋅mL^−1^ PS20 in 25 mM acetate buffer pH 5.5 at 40 °C/75% rh measured via RP-UPLC-MS were used to extract the intensities of different oxidation products. Intensities normalized to the initial content (*t* = 0) for sorbitan ethoxylated with 26 and 20 POE moieties and esterified with oleic acid (C18:1 26 POE and C18:1 20 POE), sorbitan di-oleic acid esters ethoxylated with 26 POE groups (2x C18:1), as well as oxidation products of sorbitan di-oleate (26 POE moieties) with one of the oleic acids oxidized to hydroxyl-C18:1 (C18:1 + C18:1-OH) or hydroperoxyl-C18:1 (C18:1 + C18:1-OOH) were plotted over time. The data are presented as % of the initially measured PS concentration of each formulation.

**Table 1 pharmaceutics-15-02332-t001:** Metal content analysis. ICP-MS (inductively coupled plasma-mass spectrometry) data of initially used PS20 and PS80 raw materials (RM) as well as 100 mg·mL^−1^ of PS20 and PS80 in water (UW) samples after storage for 48 weeks at 40 °C in glass vials. Additionally, 1 M acetate buffer pH 5.5 was investigated for impurities; 1 ppm = 1 ng⋅mL^−1^. Absolute concentrations; no normalization to, e.g., PS content.

Element	PS20 RM /ng⋅mL^−1^	100 mg⋅mL^−1^ PS20 UW after Storage of 48 Weeks at 40 °C /ng⋅mL^−1^	PS80 RM /ng⋅mL^−1^	100 mg⋅mL^−1^ PS80 UW after Storage of 48 Weeks at 40 °C /ng⋅mL^−1^	1 M Acetate Buffer /ng⋅mL^−1^
Al	<250	1233.5	<250	668.2	<250
B	<250	1100.5	<250	572.2	2330.5
Cu	<10	1.9	<10	12.4	<10
Fe	141.8	44.1	102.2	23.9	<100
Mo	13.4	7.4	13.8	7.4	12.7

## Data Availability

The data presented in this study are available in this article (and Appendix A).

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
