# Peer review of "Comparative Stability Study of Polysorbate 20 and Polysorbate 80 Related to Oxidative Degradation"

_pharmaceutics, 2023, doi:10.3390/pharmaceutics15092332_

Round 1

Reviewer 1 Report

In the present study, the authors have undertaken a comparative stability study to investigate the distinctions in the oxidative degradation of PS20 and PS80. They also assessed the influence of PS concentration and other factors on oxidative degradation, which holds potential relevance in the biopharmaceutical production platform. The manuscript is well-organized and carefully edited. However, several issues require attention and resolution before acceptance. These are as follows: 

1.       In the figures depicting PS content under different conditions, it is recommended to reduce the line width appropriately. This adjustment will enhance reader convenience in observing the nuances of similar parts and the overall trend of different colored lines. For instance, consider the images of 0-12 weeks at 40 °C for 438 mg · mL-1 samples. 

2.       The paper presents varying trends in PS content when different methods are employed to test PS content at certain concentrations over time. It is essential to provide a proper explanation for this phenomenon. For example, in Figure 2A and Figure 4A, where PS content changes at 100 mg · mL-1 and 438 mg · mL-1, the reason for the observed opposite trends with the prolongation of storage time should be clarified. 

3.       In the conclusion section, it would be beneficial to compare the results of this study with actual storage conditions of PS. Practical suggestions can be provided on how to address the drawbacks caused by PS oxidative degradation. Additionally, identifying aspects that warrant further investigation to mitigate PS degradation effects would be valuable. 

4.       The abstract should be revised to include all experimental variables, particularly the difference in temperature. Currently, it only highlights the differences in oxidative degradation between PS20 and PS80 and the effect of PS concentration, omitting the temperature variable. 

5.       In FIG. 2A, after 48 weeks of storage, lesser degradation of the two types of PS was observed. It is suggested to conduct supplementary pre-experiments to address this discrepancy and provide a comprehensive explanation for the observed trend. 

6.       The brief introduction of the literature experiment and the experimental conclusions in the introduction need to be revised to align with the actual conclusions of the article. Ensuring consistency in the introduction will enhance clarity and avoid any ambiguity in the presented information.

The writing of this manuscript is acceptable.

Reviewer 2 Report

The manuscript is well written with a no room for confusion in study design, methods, and results. Perhaps some parts from conclusion, particularly the assertions/reasons using other peer reviewed publications, can go under discussion section. The manuscript in general follows a logical sequence. However, I one question/suggestion to the authors: Why did the authors study only single vendor of polysorbates i.e., high purity PS20 and PS80 from Croda, but not multiple vendors? Should readers expect similar outcome for PS from other vendors while following their study design? Differences in fatty acid ester distribution are expected with different vendors as they may have their own source of vegetable oils (raw material) with different fatty acid compositions. Furthermore, their synthesis process could be different and may be proprietary. Hence we see a unique fingerprint of monoester for PS from different vendor in LCMS. When you say "only fatty acids with higher than 1.0% content we considered" for MS analysis (line #514), can the readers presume this could be applied to PS from any vendor? Perhaps adding some commentary on how your study design applies to PS from different vendor might be of interest to readers.

Minor formatting: UW abbreviation was defined multiple times. Please double check all the abbreviations and ensure they are defined only where they are first mentioned in the manuscript body.

Minor formatting: UW abbreviation was defined multiple times. Please double check all the abbreviations and ensure they are defined only where they are first mentioned in the manuscript body.

Reviewer 3 Report

I would like to express my thanks to the authors of the submitted manuscript "Comparative Stability Study of Polysorbate 20 and  Polysorbate 80 Related to Oxidative Degradation". The manuscript is timely as investigation into the degradation of polysorbates is welcome to help provide a deeper understanding into the use of polysorbates in formulating drug products and the stability impacts therein.

Keywords - there appears to be a formatting error here in the semicolon (; with unnecessary underlining.

Line 80 - A table format should not have been used for the Equations. MPDI uses a recommended style for tables.

Figure 1 -  Subscript values w , x, y and z are not clearly visible  in images A and B.

I would query  the formatting and placing of the citation markers [ ] , these should be placed inside the relevant sentence not outside like in the manuscript.

The authors should make use of paragraphs in the introduction

Section numbers should be provided.

The major gap in the research methodology is that PS20 and PS80 are rarely formulated on their own in buffered solutions  and the manuscript does not address adequately what may happen when combined with a drug product or protein. While I understand it may not be possible to carry out specific experimental work to demonstrate these effects given the variety of drug products possible but a discussion should be provided on the potential impact of changes to PS on the drug product (including proteins)

I would like to see a discussion on the effectiveness and safety of polysorbate solutions as they degrade or is the critical quality attribute simple to remain above a certain concentration/value. Is there a potential danger to patient safety with the increase in level of oxidative degradants formed during these stability studies or will these degradants catalyse degradation of api's or proteins if present.

Only minor errors in spelling, grammar, syntax and style observed. Suggestions and corrections are provided in the above section.

Round 2

Reviewer 3 Report

I would like to thank the authors for addressing my queries, I feel the changes (and in particular those requested by Reviewer 1) are sufficient to elevate the paper to merit acceptance for publication.

I would hope that the authors feel the peer review process has been helpful and brought fresh eyes to the manuscript.

There is one caveat I would like to see section numbering put in place to remain consistent with MPDI instructions for authors.

Author Response

We greatly appreciate the positive feedback regarding the revised manuscript, and we acknowledge the valuable input from the all the reviewers, which has significantly improved the manuscript's quality.

In accordance with the reviewer's request, we have implemented section numbering. We extend our gratitude for emphasizing this matter and offer our sincere apologies for not addressing it directly. The color-coding from the first revision was removed.